# Opportunities and Obstacles to the Development of Health Data Warehouses in Hospitals in France: The Recent Experience of Comprehensive Cancer Centers

**DOI:** 10.3390/ijerph20021645

**Published:** 2023-01-16

**Authors:** François Bocquet, Judith Raimbourg, Frédéric Bigot, Victor Simmet, Mario Campone, Jean-Sébastien Frenel

**Affiliations:** 1Data Factory & Analytics Department, Institut de Cancérologie de l’Ouest, 44805 Nantes-Angers, France; 2Law and Social Change Laboratory, Faculty of Law and Political Sciences, CNRS UMR 6297, Nantes University, 44313 Nantes, France; 3Oncology Department, Institut de Cancérologie de l’Ouest, 44805 Nantes-Angers, France; 4Center for Research in Cancerology and Immunology Nantes-Angers, INSERM UMR 1232, Nantes University and Angers University, 44035 Nantes-Angers, France

**Keywords:** Artificial Intelligence, Big Data Health, cancer, Comprehensive Cancer Center, French Data Protection Authority, General Data Protection Regulation, Health Data Hub, health data warehouse, hospital, oncology, National Health Data System, real-world data, real-world evidence studies

## Abstract

Big Data and Artificial Intelligence can profoundly transform medical practices, particularly in oncology. Comprehensive Cancer Centers have a major role to play in this revolution. With the purpose of advancing our knowledge and accelerating cancer research, it is urgent to make this pool of data usable through the development of robust and effective data warehouses. Through the recent experience of Comprehensive Cancer Centers in France, this article shows that, while the use of hospital data warehouses can be a source of progress by taking into account multisource, multidomain and multiscale data for the benefit of knowledge and patients, it nevertheless raises technical, organizational and legal issues that still need to be addressed. The objectives of this article are threefold: 1. to provide insight on public health stakes of development in Comprehensive Cancer Centers to manage cancer patients comprehensively; 2. to set out a challenge of structuring the data from within them; 3. to outline the legal issues of implementation to carry out real-world evidence studies. To meet objective 1, this article firstly proposed a discussion on the relevance of an integrated approach to manage cancer and the formidable tool that data warehouses represent to achieve this. To address objective 2, we carried out a literature review to screen the articles published in PubMed and Google Scholar through the end of 2022 on the use of data warehouses in French Comprehensive Cancer Centers. Seven publications dealing specifically with the issue of data structuring were selected. To achieve objective 3, we presented and commented on the main aspects of French and European legislation and regulations in the field of health data, hospital data warehouses and real-world evidence.

## 1. Introduction

If we take a look at history, the original idea of a Comprehensive Cancer Center (CCC) was developed by the British doctor William Marsden in 1851. Profoundly affected by the death of his wife from cancer, he concentrated a large part of his work on the disease and, in particular, on the classification of tumors, their causes and research into new treatments for his patients managed in a new cancer facility he created, the Royal Marsden Hospital in London. The Institute of Cancer was founded in the United Kingdom in 1909 as a research department of this hospital. In 1918 in Paris, Marie Curie and Claudius Regaud proposed a development project for the Institut du Radium, where research and therapeutic applications would be closely linked. Thus, the well-known research-to-care continuum was born, and it laid the foundation for the main concept of CCCs. In Europe, a cancer center designated as a CCC by the Organization of European Cancer Institutes (OECI) means that the center has met international and excellence standards for the following three components of its activity: cancer prevention, clinical services and research. Thus, European CCCs are at the heart of the landscape of cancer research, education and care. They are vital hubs where the historic gaps in the research-to-clinical care continuum are bridged [1]. At the end of 2022, six French cancer centers were labelled as CCCs by the OECI: the Institut de Cancérologie de l’Ouest (Angers, France), the Centre Léon Bérard (Lyon, France), the Institut Curie (Paris, France), the Institut Paoli Calmettes (Marseille, France), the Oncopole de Toulouse (France) and the Centre François Baclesse (Caen, France) [2].

Even if CCCs have established hallmarks, a greater emphasis is needed to create more effective data warehouses (DWHs) in CCCs to support the organization and the elaboration of processes essential for producing quality outcomes for patients and effectiveness in the translational process. CCC DWHs are crucial tools for improving overall understanding of cancer diseases and patient prevention, care and research in oncology worldwide [3,4,5]. Indeed, state-of-the-art DWHs that meet the highest international standards in terms of quality, interoperability and openness to outpatient settings are essential to CCCs to strengthen the relevance of the network they constitute and to deliver integrated research, outstanding innovation and excellence in patient outcomes [4]. The stakes are high here because, through the deployment of excellent DWHs, the aim is to break down silo culture and foster collaborations, leading to improved clinical research, practice changes [6] and systematized real-world evidence (RWE) studies [7], which may otherwise remain dormant or be delayed [1].

Today, the ambition of CCCs in the field of data is to respond to the challenge of the increasing complexity of patient care through the use of Big Data and Artificial Intelligence. However, let us make no mistake that, beyond the large volume of data of great diversity accumulated at high speed, the full potential of this mass of information is primarily conditioned by the ability of data producers to analyze it and then draw reliable results from it. This question, in fact, refers to two attributes of health data: its structuring (Are they structured or not? Do these data benefit from a standardized structure based on a nomenclature, a standardization, or not?) and its quality (Do the data meet quality criteria enabling us to say that they are interpretable and complete?). It is impossible to be satisfied with erroneous or fragmentary clinical data or poor quality medical imaging, which only leads to false modeling and less than robust results from Artificial Intelligence algorithms [5]. In order to meet this dual requirement of data structuring and quality, more and more CCCs have decided to develop their own DWHs [3,5,6,7,8]. The interest of these DWHs is, in particular, that they allow RWE studies to be carried out, i.e., studies conducted on the basis of data collected in routine care practice outside the traditional framework of clinical trials. Indeed, electronic health records (EHRs) linked to multiple data coming from multiscale and multisource applications (medical reports, anatomopathology, omics, etc.) are increasingly used for RWE studies in oncology in CCCs [4]. Each center has to be able to follow their cohorts of patients longitudinally and, therefore, track disease response, resistance and late effects of treatment for patients over many years, thereby building up powerful data warehouses for research and development [1]. Interaction between these DWHs and preexisting population-based databases and registries are essential developments for the coming years for CCCs, particularly for tracking the long-term medical outcomes of treatments [4]. These DWHs in CCCs can be fruitful platforms in oncology for research around survivorship, which considers cancer as a long-term condition, but also for the validation of quality-of-life indicators and health economics [3,5]. While the large-scale use of health data is a source of progress and medical advances, it legitimately raises questions of a legal and ethical nature. Because of the sensitivity of the data processed, as in other European countries, the use of hospital DWHs is subject to strict rules on the processing of patients’ personal data in France. By definition, these rules and regulations are highly country-specific. In France, the deployment of DWHs in hospitals is based on a commitment to comply with the national “Health DWH” guidelines published in 2021 by the French Data Protection Authority (CNIL—Commission Nationale de l’Informatique et des Libertés) [9]. These guidelines specify the legal framework applicable to hospital DWHs—the framework resulting from the General Data Protection Regulation (GDPR) European Law completed by some national provisions. This commitment to comply with the CNIL standards implies that French CCCs must work hard to ensure compliance with the provisions it contains.

The objectives of this article are threefold: first, to provide insight on public health stakes of developing DWHs in CCCs to manage cancer patients in a comprehensive way; secondly, to set out the challenge of structuring data through DWHs within French CCCs in order to reuse them in RWE studies; thirdly, to outline the legal issues to implement DWHs and to carry out RWE study implementation in French CCCs. To meet the first objective, this article firstly proposes to shed light on and discuss the relevance of an integrated approach to manage cancer patients in order to respond to the complexity of cancer pathology and the formidable tool that Big Data and DWHs represent to achieve this (Section 2 and Section 3). To address the second objective concerning the challenge of structuring the data in hospital oncology DWHs, we carry out a literature review on the use of DWHs in French CCCs (Section 4). Finally, to achieve the last objective, we present and comment on the main aspects of the French and European legislation and regulations in the field of health data, hospital DWHs and RWE studies (Section 5 and Section 6).

## 2. The Complexity of Cancer Pathology and the Need for a Comprehensive Approach

Cancer is a complex disease in several ways. Firstly, this pathology involves a network of dynamic interactions between an organism and its environment that evolves in time and space. As biology progresses, it is becoming increasingly clear that cancer cannot be considered as a single disease but is, in fact, a frequent nosological entity constituted as a group of rare and heterogeneous diseases [10,11].

The second major point is that the understanding of the biology of cancer is constitutive of the evolution of the classification of cancers, which, until recently, only took into account the initial tumor location and the anatomical-cytopathological analysis. The advent of molecular biology, the study of the expressions of membrane receptors on the surfaces of cancer cells and the analysis of chromosomal alterations, somatic or constitutional mutations have profoundly modified the approach to cancerous disease. Moreover, the pathophysiological mechanisms of carcinogenesis are many, leading to pathologies that are very different from one another [12]. These mechanisms include proliferation linked to autocrine growth factors and nonsensitivity to inhibitory signals; the ability to avoid apoptosis and replicate indefinitely; the ability to form metastases, genome mutations and instability; deregulation of cellular energy metabolism; the role of inflammation; and the tumor microenvironment [13].

The third essential element is the evolution of more and more cancers into chronic pathologies [4,14]. As the pathology takes a longer period of time, it is now possible to speak of a “history of the disease” made up of a succession of stages (initial diagnosis, treatments, complete or partial responses, relapses, remissions, etc.) and of health paths in cancerology. The therapeutic sequences are diversified and call upon different approaches, which are all vectors of technicization and, therefore, of increasing complexity of care, such as chemotherapy, radiotherapy, nuclear medicine, treatment by targeted therapies or immunotherapies. Added to this is the emergence of mechanisms of resistance to treatments, which give cancer pathology new properties [4,15].

The combination of these three elements, the heterogeneity of the disease, the ever-improving knowledge of the biology of cancers and their chronicization, shows the need to understand all the dimensions of cancer in an integrated approach. In contrast to historical approaches, which have tended to break down the patient care pathway by disciplines or techniques mobilized in a very segmented manner (specialized medical consultation in hospitals, conventional biological analysis in laboratories, oncology private practice in ambulatory settings, radiology, molecular biology, anatomopathology, etc.), the integrated, i.e., systemic approach seeks to integrate different levels and types of information in order to understand how this pathway actually functions. By studying the relationships and interactions between different data sources in cancer, the aim is to form a model of how the whole system works. With the advent of targeted therapies and the development of high-throughput molecular technologies that have transformed common diseases into a multitude of rare “molecular” diseases, the 1990s saw a rise in “personalized” medicine in cancer. This medicine, also referred to as “precision medicine”, can be defined as a medical approach aimed at adapting the right therapeutic strategies for the right person at the right time, determining the predisposition to diseases in the population and ensuring adapted and stratified prevention for patients.

## 3. Big Data: Digital Technology for Cancer Care and Research

Big Data health, digital health, connected health—this multiplicity of terms illustrates the scientific and technical revolution at work made possible by the use of digital technology, high-speed processing of health data, Artificial Intelligence and the profound upheavals currently taking place and yet to come in health systems. Far from being limited to facilitating the transmission of data and constituting a support to medico-technical or medico-administrative management, the concept of Big Data opens up unprecedented prospects for monitoring the state of health of populations, aiding decision making in medicine and characterizing risks [5,16].

Since the 2010s, the fields of systems biology and digital technology have gradually converged. At the heart of treatment, digital technology has made it possible to generate and exploit Big Data and to define decision algorithms that have revolutionized personalized medicine. This has naturally evolved toward an integrative and individualized approach to cancer treatment. This paradigm shift, made possible by the massification of data and their processing involving considerable computing power, has revolutionized the curative and preventive aspects of cancer care. If, outside the field of oncology, this integrated medicine is most often limited to genomic medicine, this is absolutely not the case in oncology, where it covers a much wider perimeter of data and has played an eminently more structuring role. Indeed, due to the technical nature and specialization of cancer care, many data other than those concerning the genome are considered in the treatment of cancer: those linked to RNA (transcriptomics), to proteins (proteomics), the environment in which an individual evolves (exposomics) or even radiological data (scanner, positron emission tomography, magnetic resonance imaging), which can be coupled with clinical, biological or genomic data (radiomics).

In CCCs, the stated objective is to guarantee comprehensive patient care (understanding cancer in all its dimensions: medical, biological, social, psychological, environmental, economic, etc.) that is individualized (patient-centered approach) and integrated (combining and aggregating data from heterogeneous formats and sources for the purposes of optimizing care by identifying predictive and prognostic factors for research purposes, as opposed to approaches consisting of breaking down the study of these data by discipline, such as genomics, anatomopathology, histology, radiology, etc.). For this integration of data to work and benefit patients, it is essential to strengthen the links between oncologists, surgeons, pharmacists, nurses, care assistants and biologists in a logical decompartmentalization of medical, scientific and technical disciplines. Indeed, data from care are obtained from extremely heterogeneous sources that must be aggregated and cross-referenced, which is sometimes a difficult exercise. All this illustrates the need to use massive health data to better understand cancer, improve care and optimize cancer research. This is the primary purpose of hospital DWHs.

## 4. DWH in French CCCs: The Challenge of Structuring the Data

### 4.1. The Setting for the Deployment of DWHs in CCCs in France

The National Health Data System in France is a megabase of various data whose scope includes almost all the data produced by the actors involved in the activities covered by French Health Insurance Fund. This gigantic medico-administrative database is implemented by the National Health Insurance Fund and the Health Data Hub, which is a public structure whose objective is to enable project coordinators (medical community, researchers, clinical research organizations, etc.) to easily access non-nominative data hosted on a secure platform. Although it centralizes a very wide range of data, this megabase is, nevertheless, far from exhaustive, in particular because of the fragmentation of health data in France. Indeed, clinical data are not systematically transmitted to the French Health Insurance Fund and the Health Data Hub, which makes feeding and using this national database to conduct RWE studies difficult [17]. Hospitalization data collected in CCCs are extremely rich because they concentrate several sources and are essential for understanding care pathways in oncology. These data are not exhaustively transmitted to the national database [17]. Indeed, the data transmitted to the French Health Insurance Fund are transmitted at a level of detail that is only intended to allow reimbursement by the public payer without going into the medical or technical details of the pathology or the treatment needed for implementing RWE studies. For example, pathology or genomic data are not present in the national database but only in the DWHs of the CCCs. At the same time, healthcare institutions are facing challenges in structuring and improving the quality of their data. Indeed, many of the data collected during patient health pathways are not digitalized, such as the medical reports on outpatient and previous clinical examinations [18]. The construction and use of DWHs within CCCs are carried out by health professionals and medical actors in the field who are responsible for processing these data. They, therefore, play a major role in French health Big Data and represent a valuable and unique source of clinical data in cancer care. The number of new cases of cancer diagnosed each year in France is now estimated at nearly 400,000. Three million French citizens live with a cancer that is evolving, in remission or considered cured. Every year, 160,000 people die from the consequences of cancer in France [19]. All of these patients are treated in hospitals by multidisciplinary teams, sometimes over long periods, and generate colossal volumes of data. The aim of this section is to set out the challenge of structuring data through DWHs within French CCCs in order to reuse them in RWE studies. A literature review was conducted for this purpose.

### 4.2. Methods

A literature review was performed to screen the articles published in PubMed and Google Scholar through the end of 2022. The search words included “(Oncology OR Cancerology OR Cancer) AND Data Warehouse AND France”. We selected only articles dealing specifically with the issue of data structuring within CCC DWHs.

### 4.3. Results

Seven publications were selected for the literature review. The main features and results of these publications are shown in Table 1.

### 4.4. Discussion

Due to the wide variety of data sources and the different environments in which they are produced in CCCs, health data are by nature extremely heterogeneous in terms of typology and format. The variety of data is also due to the fact that, for the same data source, the data can be in very different formats. For example, the textual data in a medical report may be in different formats or describe the same clinical situation or treatment regimen in different ways. In general, a distinction can be made between “unstructured”, “semi-structured” and “structured” data.

The first type is by far the most widespread since it represents 80% of computerized patient data in hospitals [25] and refers, for example, to textual data such as those found in hospitalization reports, consultation reports, anatomopathology reports or even multidisciplinary team meeting (MTM) reports. Another example of unstructured data is medical images. It should be noted that these unstructured imaging data may, nevertheless, be accompanied by metadata that allow the contexts in which the data are created to be understood. In the case of images, the DICOM (digital imaging and communications in medicine) standard is intended to play this role [26]. “Semi-structured” or partially structured data correspond to an intermediate type of data located between unstructured and structured data. These data can be described by characteristics that can facilitate their structuring. Technically, these are data represented in a tag-based computer language, such as XML (eXtensible Markup Language). Medical questionnaires or any other document stored in the clinical document architecture (CDA) format of the HL7 (health level 7) standard are examples of semi-structured data [27]. Finally, data are considered to be “structured” when they are formatted and transformed into a well-defined data model. Structured data are described with a repository that allows them to be enriched with semantics, thus, making their exploitation or analysis possible. This description can be standard and then shared by several data producers or local, which complicates the interoperability of the systems producing them [28], for example, in areas such as radiotherapy [20] or radiology [21].

A consubstantial element of data is its temporality. Indeed, a repeated collection of data can allow them to be represented in the form of series or chronological sequences. This is the case, for example, for biological measurements that can be carried out on patients. These data are then called signal data in the sense that they can be defined by their acquisition frequency. The notion of temporality can also embrace a wider domain, for example, in the context of reconstructing care paths. This exercise most often involves the use of unstructured data, which is not always easy in practice [29].

The aim for which the data are produced also has an impact on their characteristics and, thus, on their quality [27]. For the same information, the levels of quality requirements are not the same in the cases of a clinical trial or in routine care. Indeed, while the experimental scheme of a clinical trial provides for the collection of data within a normalized, standardized framework, the data filled in by health professionals in patient files corresponding to their routine care—so-called “real-world” data (RWD)—are often partial or incomplete. As a rule, as soon as a data source is set up for study purposes, the data are structured: data from clinical studies, registers, cohorts or even diagnosis-related froups in the medico-administrative field used for the reimbursement of care by the health insurance.

Another major element is the quality of the data contained in CCC DWHs. The quality of the data depends first of all on the purpose for which they are used and on the structure, standardization and normalization requirements linked to their use [22]. In terms of data quality, the following are commonly analyzed: missing data, duplicate data, the time required to produce data or the invalidity of data. It should be remembered that, from the perspective of secondary reuse of data, the uses are defined after the data are produced. Beyond the attributes that must be determined by the subsequent use of the data, they can, nevertheless, be judged as being of ‘sufficient’ quality if they meet the minimum criteria described by the “FAIR” principles (foundable, accessible, interoperable, reusable). Several means can be used to improve data quality to enable reuse downstream of data production, including developing quality monitoring measures throughout the data integration process to ensure that raw data are not degraded during the integration process from sources and developing analysis methods to correct data quality problems (reconciliation, deduplication, etc.). It is also possible to intervene upstream by applying corrective actions on the source applications, which is sometimes facilitated by the fact that the end users are also the data producers. Secondary reuse involves defining the dimensions of interest in terms of data quality in relation to intended uses in order to put in place indicators to assess and monitor data quality [22,23,24].

There is no doubt that CCC DWHs may rapidly become powerful tools as real-time data collection is automated and reflected in the in-hospital clinical practice for all patients hospitalized in a given hospital [17]. Furthermore, the linkage of several hospital DWHs on a wider level than a single hospital is currently ongoing including in France, as illustrated by recent initiatives [30,31,32].

### 4.5. Limitations

The literature on the subject is incomplete and remains relatively poor, as illustrated by the fact that only seven articles were selected for this review. It could be interesting to extend the review to include publications from other European countries, as well as not only to those relating to CCCs.

In the following sections (Section 5 and Section 6), we present and comment on the key elements of French and European legislation and regulations in the field of health data, hospital DWHs and RWE.

## 5. Health Data and Hospital DWHs: Legal Qualifications in France

Before analyzing the legal and the regulation aspects of the development of DWHs in CCCs in France—and to avoid any semantic misunderstanding—firstly, it is essential to qualify the concept of “health data” in the European and French sense of the term. Behind the somewhat overused terminology of “health data” lies a category of data, a generic concept that covers a wide variety of information specific to the health field. In particular, in European law, it refers to the notion of “personal data”, defined in Article 4 of the European Regulation n°2016-679 of 27 April 2016, known as the GDPR [33]. According to this regulation, “Personal data relating to the physical or mental health of a natural person, including the provision of healthcare services, […] reveal information about the health status of that person”. In French law, this terminology is also defined in Article 2 of Law No. 78-17 of 6 January 1978 in the Data Protection and Freedom of Information (DPFI) law. The text states that “Health data” means “Any information relating to an identified or identifiable natural person” (hereinafter referred to as “data subject”). The French law specifies: “An identifiable natural person is one who can be identified, directly or indirectly, in particular by reference to an identifier, such as a name, an identification number, location data, an online identifier, or to one or more factors specific to his or her physical, physiological, genetic, mental, economic, cultural or social identity” [34].

In French law, a DWH in the health field can be defined as a space in which a large volume of health data from various sources is gathered, stored for a long period and can be reused for research purposes. Although the sources of data in DWHs are varied, most of them are fed by EHRs completed by hospital health professionals and centralize a large amount of data [35]. The definition of the term warehouse is, therefore, very broad in practice. Indeed, it can cover many personal data-processing operations. The CNIL authorizes a certain number of these. The definition of the term DWH covers, in particular, a wide variety of databases whose constitution raises several issues for CCCs. Most often, DWHs are used for research in the field of health, to the extent that they are sometimes confused with research processing. In this respect, it is necessary to qualify the notion of processing precisely because research and certain DWHs are subject to separate prior formalities with the CNIL.

In order to help data controllers, the CNIL published a recommendation in 2019 specifying the criteria for determining whether a planned processing operation falls under the DWH authorization regime or, on the contrary, under the “research” authorization regime, including the volume of data collected, the purposes of the processing operation, the length of time the data are retained and the recipients of the data [36]. The details of these criteria illustrate the proximity of the DWHs to existing cohorts and registers. According to the CNIL, “The DWH collects massive data for re-use in several research projects, whereas the research responds to a precise objective and is limited in time”. Once this definition was established, the CNIL recognized that “Some projects may be more difficult to qualify (for example, a database on a specific pathology)” [36].

On the criterion of data volume, the CNIL indicates that a DWH has “massive data (data related to the patient’s medical care, socio-demographic data, data from previous research, etc.)“. Regardless of the size of a database, the data must be massive and correspond to the criteria of Big Data for velocity, speed, veracity and volume, with the latter already an established criterion. The CNIL notes that DWHs can be fed “by multiple sources (health professionals, patients, pharmacies, hospitals, etc.)” [36]. Unlike patient cohorts and registers, it is, therefore, not useful for the data controller to reuse data that it has produced itself, i.e., it is possible to feed a DWH from several sources, emanating from various structures and entities. On the other hand, data collected in the context of research are collected “specifically for research purposes” [36].

Concerning the purpose of DWHs, the CNIL considers that their main purpose is “to collect and dispose of massive data”. Thus, the CNIL considers DWHs to be a tool or a means and does not attribute a precise purpose to them. Indeed, the Commission considers that their only function is, ultimately, to gather quantitatively and qualitatively sufficient data to carry out studies, research and evaluations in the field of health. In practice, this implies that, in the eyes of the CNIL, a DWH does not in itself pursue a research purpose. On the other hand, it is the reuse of data from a warehouse for research purposes that constitutes research subject to a specific regime.

On the question of the duration of data retention, the CNIL considers that “these databases are set up for a long period (generally more than 10 years)” [36]. According to the CNIL, the length of this retention period is explained by the need to collect a substantial and sufficient amount of data for the purpose of reuse. This consideration is not neutral since it effectively excludes all temporary processing (lasting less than 10 years) from the definition of a DWH. Thus, the CNIL contrasts the duration of a research project, which is known and limited, with the duration taken into account in the case of a DWH, which is much longer.

Finally, concerning the recipients of DWHs, they are defined as recipients of personal data processing within the meaning of the GDPR [33], i.e., a natural or legal person, a public authority, a service or any other body that receives communication of personal data, whether or not it is a third party. In the case of research, the recipients are de facto restricted to the persons participating in the research. In the French sense, a DWH involves a larger list of recipients, as several research teams may need access to the data in order to, for example, analyze the data.

## 6. DWHs in French Hospitals: Strict Regulation for Their Implementation

“Seductive or frightening, data banks must increase the human capacities for information, reflection and prediction. The idea of banning them shocks the mind and would go against the trends […]. But on the other hand, to invoke the increased human power, is it not to fall into abstraction?” asked the authors of a report of the CNIL in 1975 [37]. Because of the sensitivity of health data and in the application of general rules on the processing of personal data, access to DWHs is necessarily regulated and restricted. Reconciling these requirements with the need to reuse data generates cumbersome procedures with which CCCs must comply. The DWHs in CCCs are based on a commitment to comply with the “Health DWH” guidelines drawn up by the CNIL in 2021 [9], which specified a regulatory framework resulting from the GDPR [33] and national provisions that apply to them. The use of data from DWHs for research purposes is also highly regulated by the GDPR and, for France, by the DFPI law amended accordingly in 2021 [34]. This section is not intended to provide an in-depth analysis of the CNIL framework but to discuss the main requirements it introduces for data controllers in CCCs. It then discusses the main legal and ethical imperatives related to the conduct of RWE studies from DWHs.

The guidelines published by the CNIL in late 2021 set out a number of principles relating to the governance of DWHs, the nature of the data they contain, the purposes of their processing and the procedures for accessing DWHs, with which compliance is necessary. The obligations regarding information to patients on the collection and use of their data were recalled, as were the procedures for exercising patients’ rights of access and opposition. The rules on the storage and retention of data in the warehouse were specified, as were those aimed at guaranteeing the security of personal data, whether during collection or retention. The obligations of the data controller in hospitals relating to technical and organizational measures to ensure compliance with all these rules were specified in this document [9].

Data controllers wishing to set up a DWH must comply strictly with this reference framework in order to benefit from the simplified procedure of prior declaration to the CNIL. The interest of this declaration is twofold: to avoid the more restrictive procedure of prior authorization by the CNIL and to meet the difficulty of gathering all the conditions necessary to base this processing on the consent of the data subjects. In order to be able to rely on this reference framework, it is necessary for a DWH controller to base its processing solely on the legal basis of “the performance of a task carried out in the public interest or in the exercise of official authority vested in the controller within the meaning of Article 6(1)(e) of the GDPR” [33].

Another restrictive element is that the text specified that the standard does not allow for any possible derogation from all the requirements it contains and imposes absolute and continuous compliance with each of them over time. If a CCC is unable to comply with any of the numerous requirements set out in the standard, it must apply to the CNIL for authorization in accordance with the French DPFI law. The Commission then has a period of two months, which may be extended once, to decide on the application for authorization. In the absence of a response from the CNIL within this period, the application is deemed to be favorable. It should be noted that the Commission states that it advises organizations applying for authorization to draw up a document identifying and justifying any deviations in processing operation from the guidelines [9].

Lastly, DWHs whose processing is based on the consent of the data subjects (Article 9.2. a) of the GDPR) are, in fact, excluded from the specific provisions of the DFPI law relating to the processing of personal data (Article 65 of the DFPI law). In this case, a data controller does have to carry out any specific formality with the CNIL. It is useful to underline that, in order to base a processing operation on the consent of data subjects, a data controller must comply with all the conditions for the validity of consent. Indeed, consent must be a manifestation of “free, specific, informed and unambiguous” will (Article 4.11 of the GDPR) by which a data subject agrees to the processing of his or her data. Consent must also meet the conditions laid down in Article 7 of the GDPR, namely, it must be presented in a form that is comprehensible, easily accessible and can be withdrawn at any time. In practice, the conditions of validity of consent and the possibility for a data subject to withdraw it at any time limit the interest of this basis for carrying out a DWH.

## 7. RWE Studies: What Are the Legal and Ethical Requirements in France?

In French law, research involving humans is interventional or observational research for the development of biological or medical knowledge that requires the active participation of individuals, e.g., research on the evaluation of a health product. In contrast, research not involving humans—improperly called RWE—does not require the active participation of individuals, since it involves the reuse of data generated during care, such as the data contained in a DWH. These RWE studies are subject to very strict supervision by the GDPR [33] and by the DPFI law [34], which have been amended accordingly.

Firstly, it is essential to know that pseudonymized data from DWHs are legally classified as personal data and must, therefore, comply with the regulations inherent in such data. According to European law, pseudonymization is processing of personal data in such a way that it is no longer possible to attribute data relating to a natural person without further information [33]. It is one of the measures recommended by the GDPR to limit the risks associated with the processing of personal data. It should be noted that pseudonymization is not, however, a method of anonymization. Indeed, unlike anonymization, it is reversible. Pseudonymization is intended to reduce the risk of correlating a set of data with the identity of a person, but it is not a foolproof method because pseudonymized data can still be reidentified indirectly by cross-referencing information. This is why pseudonymized data are legally considered personal data in Europe.

The second important element to consider is that, while European and French regulations lay down a principle of prohibition on the processing of sensitive data, which form a special category of personal data (Article 9 of the GDPR) that includes health data, there are, nevertheless, exceptions. Indeed, it remains possible to process sensitive data if a person has given his or her consent (Article 9-2 a) GDPR) [31]; if the processing of data for scientific research purposes can be considered proportionate to the objective pursued and presents appropriate safeguards for the rights and freedoms of a data subject (Article 9-2 j GDPR) [31]; or if the processing is necessary for reasons of public interest in the field of public health, such as guaranteeing high standards of quality and safety of health care or the use of a medicine or medical device. In this respect, it should be noted that, in France, data processing in research can only be carried out in consideration of the public interest it presents, as provided by the DFPI law [34]. Compliance with these measures takes the form of an authorization from the CNIL or a commitment to comply with one of the reference methodologies approved by the Commission.

Despite the fact that the French DFPI law has been in force since 1978, it is only since the 2000s and the appearance of the notion of “personal data” that any health research using personal data must obtain authorization or undertake to comply with a reference methodology approved by the CNIL. Since 2018, which corresponds to the date on which the RGPD came into force, research on data has been governed simultaneously by the European regulation and the DFPI law [34]. This legal and regulatory framework aims to ensure the protection of individuals by protecting patients’ personal data and, in particular, to ensure that patients cannot be identified directly or indirectly through their personal data. For RWE studies, three prerequisites are imposed: patients must be informed of any research reusing their personal data and must not object to it; reused data must be limited to the strict needs of the research (collection of unnecessary data is prohibited); and reused data must be coded and secured, i.e., pseudonymized or anonymized. In addition, projects using DWH data must be authorized either by a scientific council including a member qualified in ethics or by a local ethics committee. As for patients, they must be informed of the integration of their data in a DWH and be able to object at any time or give their consent.

## 8. Conclusions

In this article, we firstly underlined the absolute necessity of having an integrated and comprehensive approach to manage cancer patients, as well as interest in implementing robust DWHs to respond to the complexity of cancer disease in CCCs. Secondly, the literature review that we conducted allowed us to illustrate the inherent difficulty of data structuring in these DWHs in French CCCs given their heterogeneity and their multisource, multidomain and multi-scale nature. Finally, through a discussion on key elements of French and European legislation and regulations in the field of health data, hospital DWHs and RWE, we presented the main legal and regulatory challenges of deploying these warehouses. The first DHW development initiatives in French CCCs represent a complete paradigm shift for health professionals. Indeed, the data become the central subject in the sense that an approach followed with massive data is not based on pre-existing information structures but on the data themselves, whatever their nature. The traditional approach in medicine has been based on the following sequence: information (or hypothesis), followed by data and then knowledge. The approach with DWHs breaks down as follows: data, followed by information (or hypothesis) and then knowledge. While reasoning in medicine used to be deductive, it is now inductive, which is a revolution in itself [13]. It is now appropriate to extend this study to other countries and other types of hospitals treating cancer patients.

## Figures and Tables

**Table 1 ijerph-20-01645-t001:** Publications selected for the literature review on data structuring in CCC DWHs.

Authors	Dates	Aims and Methods	Main Findings
Lauzanne et al. [6]	2022	To use open government data in France from 1970 to 2021 to identify deceased patients and match them with patient data collected from the Institut de Cancérologie de l’Ouest DWH between 2015 and 2021.	To evaluate the algorithms, a deterministic record linkage was performed for an exact matching algorithm and a fuzzy matching algorithm. The exact matching algorithm allowed doubling the number of dates of death in the hospital DWH, and the fuzzy matching algorithm tripled it.
Casarotto E et al. [17]	2020	To identify databases to study RWD of anticancer drugs in France by reviewing the literature.	Large medical record databases were lacking, but efforts were made to give access to hospital DWHs for RWE studies. Databases resulting from ad hoc collections were available for some cancer localizations and allowed obtaining highly valuable clinical and biological data.
Bibault JE et al. [20]	2018	To develop ontology in radiation oncology to address the issue of the lack of semantic interoperability.	The new ontology reported all anatomical and treatment-planning structures and could be used to integrate dosimetric data in the Paris public hospital DWHs.
Lemordant et al. [21]	2022	To integrate clinical image data in a hospital DWH via a web service to query and access pixel data from a DWH (prostate cancer use case).	All patients and imaging tests of the prostate cancer cohort were retrieved via a scalable and domain-neutral approach with a precision of 0.95 and a recall of 1.
Tsopra R et al. [22]	2021	To develop a European framework for assessing AI for predicting treatment response in TNBC using RWD and molecular omics data from DWHs and biobanks.	This framework was based on 7 steps and formed the basis of a validation platform. This made it possible to assess and compare AI algorithms for predicting the response to TNBC treatments with external RWE datasets.
Ansoborlo M et al. [23]	2021	To develop an NLP automated algorithm to detect patient and tumor characteristics in MTM for lung cancers reports to reduce the time-consuming prescreening for trial inclusions in oncology.	The performance parameters of the algorithm were a macroaverage F1-score of 93%, a precision of 98% and a recall of 92%. In MTM, fill rate variabilities among patient and tumor information remained important (from 31.4% to 100%). The most difficult to automatically collect were genetic mutations and rearrangement test results.
Zapletal et al. [24]	2022	To use radiation therapy data in projects related to rectal cancer patients and assess the feasibility of integrating radiation oncology data into a hospital DWH.	A pipeline to integrate radiation therapy data into the Georges Pompidou European Hospital i2b2 instance was developed and evaluated with a cohort of 262 patients.

AI: Artificial Intelligence; CCC: Comprehensive Cancer Center; DWH: data warehouse; i2b2: Informatics for Integrating Biology and the Bedside; MTM: multidisciplinary team meeting; RWD: real-world data; RWE: real-world evidence; TNBC: triple-negative breast cancer.

## Data Availability

No research data or datasets were created for this research.

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
