# Peer review of "Opportunities and Obstacles to the Development of Health Data Warehouses in Hospitals in France: The Recent Experience of Comprehensive Cancer Centers"

_ijerph, 2023, doi:10.3390/ijerph20021645_

Round 1
Reviewer 1 Report
Article Type: Literature Review
Full Title: Data Warehouses and Real-World Evidence Studies in French Comprehensive Cancer Centers: Public Health, Data Structuring and Legal Issues
The present study aimed to investigate the case of Comprehensive Cancer Centers (CCCs) in France through literature analysis. This review study aimed to provide insight into the public health stakes of developing DWHs in CCCs to manage cancer patients comprehensively; to set out the challenge of structuring the data through DWHs within French CCCs to re-use them in RWE studies; to outline the legal issues to implement DWHs and to carry out RWE studies implementation in French CCCs. This is a timely and promising study. Nevertheless, the study has some methodological weaknesses that need to be addressed.
The abstract should provide more details about the methodology (data collection, data analysis, etc.). The abstract should mention the aim, setting and design, materials and methods, statistical analysis used, and results.
Even if this is a literature review study, a specific section for the “Method” is needed. This section should present which method was used in selecting and presenting the articles. How the articles related to the subject were obtained? Which databases (such as WoS, Scopus, and Google Scholar) were utilized? What were the criteria to include the articles? Basic information about each article examined in tabular form (who made it, when, by what method, and reached the findings) should be presented.
Reviewer 2 Report
The manuscript is written in a very structured, clear, and coherent manner. The synergy between healthcare, information storage, and French law is presented. However, the content of the research raises several clarifying questions.
1. Do not use acronyms in the abstract. You enter them precisely in the text of the manuscript later.
2. Having such DWHs and data mining can help medical and health specialists to discover knowledge on healthcare, institutional planning, optimization and management. This is the case not only in France, but also in other countries, such as other EU states or the USA.
· Kraujalis, et al. Mortality rate estimation models for patients with prostate cancer diagnosis. Baltic journal of modern computing. 2022, vol. 10, no. 2, p. 170-184. https://doi.org/10.22364/bjmc.2022.10.2.06.
Expand the introduction with it.
3. The conclusion does not need to review the abstract philosophy of data storage. Instead, the conclusion is a concise summary of the final results of the entire work. Therefore, the most important statements are formulated here.
Round 2
Reviewer 1 Report
Authors have addressed all of my concerns.